# Cerebral Benefits Induced by Electrical Muscle Stimulation: Evidence from a Human and Rat Study

**DOI:** 10.3390/ijms25031883

**Published:** 2024-02-04

**Authors:** Rémi Chaney, Clémence Leger, Julien Wirtz, Estelle Fontanier, Alexandre Méloux, Aurore Quirié, Alain Martin, Anne Prigent-Tessier, Philippe Garnier

**Affiliations:** 1INSERM UMR1093-CAPS, Université Bourgogne Franche-Comté, UFR des Sciences de Santé, F-21000 Dijon, France; chaney-52@hotmail.fr (R.C.); clemence.leger@u-bourgogne.fr (C.L.); julien.wirtz@u-bourgogne.fr (J.W.); estelle.fontanier@u-bourgogne.fr (E.F.); alexandre.meloux@u-bourgogne.fr (A.M.); aurore.quirie@u-bourgogne.fr (A.Q.); pgarnier@u-bourgogne.fr (P.G.); 2INSERM UMR1093-CAPS, Université Bourgogne Franche-Comté, UFR des Sciences du Sport, F-21000 Dijon, France; alain.martin@u-bourgogne.fr; 3Département Génie Biologique, IUT, F-21000 Dijon, France

**Keywords:** electrical muscle stimulation, brain-derived neurotrophic factor, lactate, FNDC5/Irisin, muscle-brain crosstalk, cognition

## Abstract

Physical exercise (EX) is well established for its positive impact on brain health. However, conventional EX may not be feasible for certain individuals. In this regard, this study explores electromyostimulation (EMS) as a potential alternative for enhancing cognitive function. Conducted on both human participants and rats, the study involved two sessions of EMS applied to the quadriceps with a duration of 30 min at one-week intervals. The human subjects experienced assessments of cognition and mood, while the rats underwent histological and biochemical analyses on the prefrontal cortex, hippocampus, and quadriceps. Our findings indicated that EMS enhanced executive functions and reduced anxiety in humans. In parallel, our results from the animal studies revealed an elevation in brain-derived neurotrophic factor (BDNF), specifically in the hippocampus. Intriguingly, this increase was not associated with heightened neuronal activity or cerebral hemodynamics; instead, our data point towards a humoral interaction from muscle to brain. While no evidence of increased muscle and circulating BDNF or FNDC5/irisin pathways could be found, our data highlight lactate as a bridging signaling molecule of the muscle–brain crosstalk following EMS. In conclusion, our results suggest that EMS could be an effective alternative to conventional EX for enhancing both brain health and cognitive function.

## 1. Introduction

Physical exercise (EX) has emerged as a potent modulator of brain health, exerting pronounced effects on cognitive functions, particularly executive functions, memory, and well-being, by reducing stress and enhancing mood [1,2]. Investigations conducted in rodent models have firmly established the pivotal role of brain-derived neurotrophic factor (BDNF), a neurotrophin governing synaptic plasticity, neurogenesis, and neuronal survival, in mediating the positive impacts of EX on the brain [3,4,5]. Similar significance has also been reported in humans, since the val66met polymorphism (single-nucleotide polymorphism in the *bdnf* gene corresponding to a valine-to-methionine substitution), which is associated with a defect in activity-dependent regulated secretion, attenuates the cognitive advantage of EX [6,7,8]. Three main mechanisms have been proposed to explain the EX-induced brain BDNF overproduction: an increase in its neuronal expression through an activity-dependent mechanism, a contribution of hemodynamics through cerebral blood flow (CBF) elevation, and recently, a process involving the capacity of contracting muscles to synthesize and secrete molecules termed “myokines” into the bloodstream [9,10,11]. Remarkably, some of these myokines possess the ability to cross the blood–brain barrier (BBB), thereby facilitating an increase in cerebral BDNF production.

Among these myokines, irisin and lactate have garnered sustained attention in recent years as mediators of EX-induced effects on cerebral health. Initially discovered for its critical role in regulating energy metabolism through the conversion of white adipose tissue to brown adipose tissue, irisin is regulated by the peroxisome proliferator activator receptor γ coactivator-1α (PGC-1α) pathway, cleaved from fibronectin type III domain-containing protein 5 (FNDC5) and released in the blood circulation [12]. Recent research provided compelling evidence suggesting that the activation of the PGC-1α/FNDC5/irisin pathway within skeletal muscles is also involved in EX-induced cerebral plasticity [13,14]. In addition, recent discoveries confirmed the ability of irisin to cross the BBB and/or bind to integrin receptors, subsequently triggering several intracellular pathways that contribute to its neuroprotective effects [14,15]. Concerning lactate, this metabolite is generated in response to anaerobic glycolysis activation in skeletal muscles and was shown to cross the BBB through monocarboxylate transporter (MCT) [16]. In rodents, lactate participates in neuroplastic processes such as neurogenesis [17], neuronal excitability [18], and long-term potentiation (LTP) [19]. Interestingly, using the intraperitoneal injection of a lactate MCT inhibitor in mice submitted to voluntary EX, a study reported that hippocampal *bdnf* gene expression was abolished [20]. Of note, in humans, the diffusion of lactate within the cerebral parenchyma was positively linked to improvements in executive functions, as assessed through the Stroop task [21].

Despite the compelling evidence and prevailing recommendations underscoring EX potential to enhance cerebral function, numerous individuals encounter obstacles when attempting to engage in active physical activity. For these individuals, sessions of electromyostimulation (EMS), also known as neuromuscular electrical stimulation (NMES), which involve inducing involuntary muscle contractions using an external source of current, are used as a passive substitute for EX [22]. Indeed, the application of EMS on the quadriceps muscle has proven effective in mitigating muscle atrophy, improving the cardiorespiratory function, and enhancing carbohydrate metabolism [23,24,25,26]. However, the impact of EMS on brain health has been relatively unexplored. In this context, we assessed the effect of EMS on cognition in humans and on the expression of BDNF in regions related to cognition in rats (prefrontal cortex, hippocampus), as well as the underlying mechanisms, involving neuronal activity, cerebral shear stress, and the release of BDNF, irisin, and lactate by skeletal muscle.

For this purpose, both Wistar rats and human participants underwent two identical sessions of EMS, spaced a week apart. Each session involved a 30 min period of EMS applied to the quadriceps muscles. The stimulation followed a specific protocol, including a frequency of 100 Hz, a pulse duration of 400 µs, an ON/OFF ratio of 7 s/14 s, and a gradual intensity increase from 6 mA to 20 mA in rats, and from 6 mA to the maximum tolerable intensity in humans (48.9 ± 22.4 mA). For the human experiment, cognitive and mood assessments were conducted immediately and 24 h after the second EMS session using the Stroop task, the Rey figure, a 15-word list, and the Profile of Mood States (POMS) questionnaire. In rats, tissue samples from the prefrontal cortex, hippocampus, and quadriceps muscles were collected either 4 h or 24 h post-EMS for histological and biochemical analyses, including Western blotting and RT-qPCR. The circulating levels of BDNF and irisin were measured by ELISA immediately, 4 h, and 24 h post-EMS, while lactatemia was measured using reactive bands in humans and rats immediately at the end of the second EMS session.

## 2. Results

### 2.1. Characterization of the EMS Protocol in Humans

We conducted a study with 40 healthy subjects, applying, by means of three electrodes positioned on the right femoris, the vastus lateralis, and the vastus medialis of the quadriceps (Figure 1A), a high-frequency electrical current (100 Hz), with a 7 s contraction duration alternating with a 14 s rest period, for 30 min, increasing the intensity from 6 mA to the maximal tolerable level (48.9 ± 22.4 mA). These subjects were randomly assigned to two groups: a control group (CTRL, n = 20) and an EMS group (n = 20). As already reported [27,28], the EMS protocol resulted in a decrease in maximum voluntary force (MVC), measured immediately after the familiarization session (EMS 1, −32.4 N.m, *t*-test, *p* = 0.0011, Figure 1B) as well as during the assessment session (EMS 2, −49.6 N.m, *t*-test, *p* < 0.0001, Figure 1B). Importantly, the loss of MVC between EMS 1 and EMS 2 was not statistically different. Furthermore, the EMS protocol induced significant muscle soreness during each session, measured using a pain scale ranging from 0 to 10 (ANOVA, time factor, F(6, 168) = 359.8, *p* < 0.0001, session factor, F(1, 168) = 427.3, *p* < 0.0001, interaction, F(6, 168) = 54.02, *p* < 0.0001, Figure 1C). Specifically, the peak of muscle soreness was observed two days after EMS for the familiarization session, while it was noted one day after EMS for the second session. Moreover, although muscle soreness was present (>1) five days after EMS 1, it almost disappeared (<1) four days after EMS 2. Finally, it is noteworthy that muscle soreness was less pronounced during EMS 2 compared to EMS 1 (*p* < 0.0001, Figure 1C).

Throughout the 30 min EMS protocol, we measured heart rate (HR), blood pressure (BP), oxygen saturation (SaO_2_), and frontal temperature (T°) (Figure 1D–G). In the CTRL group, no significant changes in HR, BP, and SaO_2_ were observed during the 30 min of the experiment. However, during the last 10 min of the EMS session, we observed a slight significant increase in HR from 5.9 beats per minute (bpm) to 8.1 bpm compared to the baseline values (ANOVA, time factor, F(6, 266) = 10.06, *p* < 0.0001, group factor, F(1, 266) = 12.57, *p* = 0.0005, interaction, F(6, 266) = 9.802, *p* < 0.0001, Figure 1D), with no notable variation in BP (Figure 1E) and a small decrease in SaO_2_ ranging from −2.07% to −2.97% (ANOVA, time factor, F(6, 266) = 16.13, *p* < 0.0001, group factor, F(1, 266) = 305.5, *p* < 0.0001, interaction, F(6, 266) = 14.90, *p* < 0.0001, Figure 1F). Regarding T°, the EMS group maintained a stable temperature throughout the 30 min session, while the CTRL group showed a very weak decrease of −0.15 °C during the last 10 min (ANOVA, time factor, F(6, 266) = 27.23, *p* < 0.0001, group factor, F(1, 266) = 4.575, *p* = 0.0334, interaction, F(6, 266) = 7.221, *p* < 0.0001, Figure 1G).

### 2.2. Effect of the EMS Protocol on Cognitive Function and Mood in Humans

To assess the effects of EMS on cognitive function and mood, each subject underwent two sessions between 5:00 p.m. and 6:00 p.m. (Figure 2A). During the first session, the participants completed a cognitive test (Stroop task) and a mood questionnaire (POMS) before the EMS session to familiarize them with the EMS procedure and the cognitive tests. Only the EMS group underwent the EMS familiarization. After a 7-day interval, the subjects underwent a PRE-TEST (Stroop task, POMS), followed by an EMS session or 30 min of rest and ending with a POST-TEST (Stroop task, POMS, 15-word list, Rey figure). For the 15-word list, a recall was also performed 24 h following EMS.

EMS led to more significant improvements than the CTRL condition in both congruent (+4.55 words, *t*-test, *p* = 0.0059, Figure 2B) and incongruent conditions (+3.1 words, *t*-test, *p* = 0.0457, Figure 2D). Indeed, The CTRL task did not impact the performance in both the congruent (−0.5 word) and the incongruent conditions (+1.2 word). However, in the neutral condition, EMS and CTRL task had the same effect on performance (+2.6 and +1.8 words respectively, Figure 2C). The number of errors was not different between the groups. Regarding the memory tasks, there was no change in the ability to reproduce the Rey figure (Figure 2E) or to recall the list of words between the two groups (Figure 2F).

Regarding the POMS, a decrease in anxiety was observed in the EMS group compared to the control group (*t*-test, *p* = 0.0007, Figure 2G). Interestingly, the effect of EMS on the anxiety scores was also observed during the familiarization session. As for other mood states such as anger, fatigue, depression, confusion, and vigor, no significant change was observed in either group (Figure 2H–L). Nevertheless, these results demonstrate that acute EMS has the potential to improve executive function and reduce anxiety in humans.

### 2.3. Characterization of the EMS Protocol in Rats

We developed and characterized the EMS protocol in rats, replicating the same quadriceps-applied EMS protocol in humans. Therefore, in rats previously anesthetized with isoflurane, the hind limbs were shaved to position two electrodes on the skin overlying the quadriceps muscle (Figure 3A). The intensity was set at 6 mA at the beginning of the session and gradually increased to 20 mA. After the procedure, the animals were awakened and returned to their respective cages. Interestingly, the wake-up time for the electro-stimulated animals was significantly reduced compared to that for the SHAM animals (−224 s, *t*-test, *p* < 0.0001, Figure 3B). To acclimate the animals to the experimental protocol and EMS, an initial EMS familiarization session was conducted 7 days after their arrival in the laboratory. Subsequently, 7 days later, the animals received a second EMS application and were sacrificed either 4 or 24 h after this session (Figure 3C).

We monitored the body weight of the animals throughout the experimental procedure (day 0 to 14) and did not observe any significant difference between the SHAM and the EMS groups for both the 4 h (Figure 3E) and the 24 h sacrificed cohorts (Figure 3E), suggesting that our experimental protocol did not induce stress, affect cage locomotion, or alter food intake. Similarly, there was no difference in weight or visual appearance of the quadriceps muscle between the two groups (Figure 3F,G). Since it is documented in the literature that EMS can induce alterations in muscle tissue [29], manifested by sarcomere tearing, sarcolemma permeabilization, and sometimes muscle fiber death, we conducted H&E staining of the quadriceps muscles to assess the potential infiltration of inflammatory cells. As previously described in humans, and with the purpose of phagocytosing damaged cells and debris and secreting mediating molecules capable of inducing the proliferation and differentiation of satellite cells for the repair of muscle tissue, we observed a rare infiltration of inflammatory cells (Figure 3H). As apoptosis has been detected in muscular diseases and is involved in myofiber cell death, we also assessed caspase 3 and cleaved caspase-3 expression [30]. The modest alteration in the muscle tissue was confirmed by the absence of an increase in caspase-3 expression and the undetectable levels of its cleaved form (activated) in quadriceps muscle tissue at the 24 h time point (Figure 3I).

### 2.4. Effect of the EMS Protocol on BDNF and Synaptic Protein Expression in Rats

We examined the impact of the EMS protocol on BDNF expression in the prefrontal cortex and hippocampus. We did not observe any changes in bdnf mRNA expression in the prefrontal cortex 4 h after EMS, as assessed by RT-qPCR (Figure 4A). Consistent with these results, immunoblot analysis revealed no change in BDNF protein expression in the prefrontal cortex either at 4 or 24 h after the EMS session (Figure 4B). In contrast, we found a significant elevation in bdnf mRNA levels in the hippocampus 4 h after EMS (+116%, *p* = 0.0079, *t*-test, Figure 4C). To confirm that the increase in bdnf mRNA was associated with an elevation in the protein levels, we conducted immunoblots. The results showed that 24 h after the EMS session, BDNF protein expression was significantly increased (+240%, *t*-test *p* < 0.0001, Figure 4D), with no change observed at 4 h.

BDNF plays a crucial role in synaptic physiology by enhancing baseline synaptic transmission, promoting LTP, and facilitating synaptogenesis. Therefore, at the 24 h time point, aligning with the concurrent increase in BDNF protein expression, we investigated the expression of several synaptic proteins, including post-synaptic density 95 (PSD-95), growth-associated protein 43 (GAP-43), and synaptophysin (SYP), in the hippocampus of SHAM and EMS rats. We observed an increase in the hippocampal expression of PSD-95 (+27%, *t*-test, *p* = 0.0414) and GAP-43 (+47%, *t*-test, *p* = 0.0067) and no change in SYP expression (Figure 4E).

### 2.5. Effect of the EMS Protocol on Neuronal Activity and Hemodynamics in Rat

After observing that EMS induced an elevation of the BDNF level in the rat hippocampus and behavioral improvements in humans, we sought to better understand the underlying mechanisms of these effects. Therefore, we assessed the hippocampal expression of c-fos and p-eNOS^Ser1177^ as respective markers of neuronal and hemodynamic pathways in the SHAM and EMS groups. However, we did not find any variation in the hippocampal expression of c-fos and p-eNOS^Ser1177^ 4 h and 24 h after EMS (Figure 5A,B). These results suggest that following EMS, the increase in BDNF is independent of neuronal and hemodynamic pathways and is rather related to other mechanisms, such as the muscle–brain crosstalk.

### 2.6. Effect of the EMS Protocol on Skeletal Muscle and Circulating BDNF in Rats

To explore the humoral pathway, since skeletal muscle was shown to be a potential source of BDNF [31,32], we first investigated whether muscle could produce and secrete BDNF into the circulation and contribute to the elevation of the hippocampal BDNF levels in response to EMS. Our analysis revealed no significant change in BDNF protein expression in the quadriceps muscle, either 4 h or 24 h after EMS (Figure 6A). Furthermore, no variation in the serum levels of BDNF was observed immediately (blood collected from the tail, Figure 6B) or 4 and 24 h (blood collected intracardially, Figure 6C) after EMS, as assessed by ELISA. Taken together, these data indicate that muscle-derived BDNF does not appear to act as a myokine in our model of EMS.

### 2.7. Effect of the EMS Protocol on the FNDC5/Irisin Pathway in Rats

Subsequently, we turned our attention to irisin, one of the most studied myokines in the context of the muscle–brain crosstalk. Given that irisin is cleaved from FNDC5, it is crucial to note that immunological approaches such as immunoblotting detect both FNDC5 and irisin. Furthermore, the molecular weight of irisin is approximately 24 kDa due to dimerization and glycosylation mechanisms, which also corresponds to the molecular weight of FNDC5 [33]. Thus, when measured in tissues, the terminology here used was FNDC5/irisin, in accordance with the literature [13]. In our Western blot of quadriceps and hippocampus homogenates, we found a band at 24 kDa. However, as FNDC5 is a transmembrane protein, it should not normally be found in the circulation. Therefore, for the assessments in the circulation, we specifically refer to irisin.

We demonstrated that EMS led to a significant increase in the expression of FNDC5/irisin in the quadriceps muscle 24 h after EMS (+110%, *t*-test, *p* = 0.0466, Figure 7A), with no observed change at 4 h. However, the expression of FNDC5/irisin in skeletal muscle was not correlated with hippocampal BDNF expression 24 h after the EMS sessions (Figure 7B).

Surprisingly, despite an elevation in the muscle expression of FNDC5/irisin, we did not observe an increase in the circulating irisin levels 0, 4, or 24 h after the EMS session, using an ELISA approach with one of the most used kits in the literature (EK-067-29, Phoenix Pharmaceuticals, Figure 7C,D). Since the ELISA kits used to detect circulating irisin have been criticized for their cross-reactivity with other serum proteins, we performed an immunoprecipitation followed by Western blotting on rat serum pools from the SHAM and EMS groups. We identified a band at around 28 kDa in the immunoprecipitated serum, which could correspond to a glycosylated form of irisin (Figure 7E). However, we did not observe a significant variation in the intensity of this band between the EMS and the SHAM groups. In fact, we even noticed a trend towards a decrease in the serum irisin levels collected immediately and 4 h after EMS (Figure 7E).

Finally, since it was demonstrated that irisin can cross the BBB and bind to the αVβ5 integrin receptor [14,34], thereby triggering focal adhesion kinase (FAK)-dependent signaling cascades (Figure 7F), we measured the hippocampal expression of FNDC5/irisin (Figure 7G) and p-FAK^Tyr397^ (Figure 7H) in the SHAM and EMS groups at both 4 and 24 h. No variation could be observed in the EMS group compared to the SHAM animals. In summary, our results suggest that the increase in brain BDNF expression in response to EMS is independent of the peripheral and central productions of FNDC5/irisin.

### 2.8. Effect of the EMS Protocol on Blood Lactate Release in Rats and Humans

Importantly, we observed that our quadriceps-applied EMS protocol increased the blood lactate levels similarly in rats (+323%, *t*-test, *p* < 0.0001, Figure 8A) and humans (+339%, *t*-test, *p* = 0.0108, Figure 8B). It was reported that lactate secreted by skeletal muscles during EX can contribute to the elevation of the hippocampal BDNF level in a SIRT1-dependent manner. SIRT1 is a NAD^+^-dependent deacetylase capable of modulating the activity of several transcription factors [35,36]. Here, we show that EMS resulted in an elevation of SIRT1 expression 4 h after stimulation (+42%, *t*-test, *p* = 0.0410, Figure 8C), suggesting the involvement of the lactate/SIRT1 pathway in the increase in cerebral BDNF levels in response to EMS.

Consistent with this hypothesis, we found a positive correlation between lactate production in rats in response to EMS and BDNF protein expression in the hippocampus 24 h after the application of the experimental protocol (Pearson correlation, r = 0.8891, *p* < 0.0001, Figure 8D), while in humans, lactate production was inversely correlated with the decrease in anxiety scores induced by EMS (Pearson correlation, r = 0.8815, *p* < 0.0038, Figure 8E). Although our results did not reveal a significant correlation, a trend for a positive association was observed between blood lactate levels and improvement in the congruent (Pearson correlation, r = 0.6398, *p* < 0.0875, Figure 8F) and incongruent conditions (Pearson correlation, r = 0.6468, *p* < 0.0830, Figure 8G) of the Stroop task.

## 3. Discussion

The purpose of our study was twofold. Firstly, we aimed to evaluate the impact of an EMS protocol on cognitive function in humans and BDNF-dependent brain plasticity in animals as an alternative to classical EX. Secondly, we sought to better characterize the determinants of the muscle–brain dialogue, since our EMS protocol applied to the quadriceps enabled us to isolate muscle contraction from systemic responses, such as increased neuronal activity, cerebral blood flow, and release of exerkines by organs other than the muscle, which are induced following a conventional EX protocol [5,37].

Our findings demonstrate that EMS in humans can enhance executive function, as evaluated by the Stroop task, without affecting the mnemonic faculty assessed by the Rey complex figure and a list of 15 words. Regarding mood, we observed a reduction in the anxiety scores in response to the EMS protocol. Although significant but modest compared to the effect of classical exercise, the cognitive benefits observed in humans were corroborated with a selective upregulation of BDNF and synaptic proteins in the rat hippocampus. We demonstrated that the elevation of hippocampal BDNF in rats in response to EMS was independent of an increase in neuronal activity and cerebral hemodynamics, as well as of the production of the myokines BDNF and FNDC5/irisin. Conversely, the behavioral benefits observed in humans and the elevation of BDNF in rats were positively correlated with lactatemia, suggesting that lactate could represent an appealing candidate to explain the changes in hippocampal BDNF expression in response to the EMS protocol.

To our knowledge, our study is the first to demonstrate a positive effect of EMS on anxiety and executive function in humans, showing a significant improvement in the congruent and incongruent conditions of the Stroop task. While previous studies using 30 min EMS failed to report such an impact using various tests (the Stroop task, the Go/No Go task, and the Wisconsin Card Sorting Test) [38,39], this discrepancy could be attributed to the frequency used in these studies, since the EMS protocols were applied using a very low frequency (4 Hz) that induces muscle contractions perceived as mild sensations of skin tingling. In contrast, we opted for a high-frequency stimulation protocol (100 Hz) inducing tetanic muscle contractions, because these protocols have been reported to be efficient to improve muscle strength and mass [40,41]. In addition, although our protocol induced muscle soreness, this was reduced after the second session and offered the advantage of a maximal muscle contraction. Although not measured in our studies, it is plausible that the difference in sensory perception between low and high-frequency EMS protocols leads to different modifications in arousal, with a “relaxing” effect from muscle twitches induced by low frequencies and a “stimulating” effect from tetanic contractions induced by high frequencies. Consistently, it was suggested that acute EX may promote cognitive improvements by influencing the arousal system in an intensity-dependent manner [42,43]. Compared to conventional EX, our results demonstrate that EMS only partially replicates the effect of EX. Indeed, a high-intensity EX protocol was reported to enhance the Stroop task in all conditions and to affect more mood aspects [44], as, in addition to anxiety that was reduced in our EMS experimental conditions, depression, confusion, and anger were also reduced after acute EX [2]. These discrepancies are most likely due to the difference in muscular mass involved when comparing EMS to conventional EX, as well as to the mechanisms triggered in response to these two different paradigms. Thus, HR exhibited only a marginal increase following EMS, whereas conventional EX, contingent on intensity, was documented to induce a significant elevation in HR [45]. In terms of BP, no significant change was observed during the EMS protocols, while it is established that EX elevates BP by up to 30% [44]. Furthermore, while conventional EX was reported to potentially decrease SaO_2_ by over 5% [46], our study indicated that EMS resulted in a SaO_2_ 2% reduction of only 2%. Future within-study comparisons are necessary to better understand the impact of stimulation parameters on cognitive function, notably to optimize EMS protocols for chronic applications.

As stated in the introduction, a brain BDNF level increase can be considered a solid index of cognitive function, since its critical role was demonstrated in animals and humans in the context of EX. Using the same EMS protocol in animals and humans, we aimed at delineating the molecular mechanisms involved focusing on this neurotrophin in two brain regions engaged in cognitive functions, i.e., prefrontal cortex and hippocampus. We demonstrated that EMS had no effect on the prefrontal region but resulted in an upregulation of *bdnf* mRNA and BDNF protein expression at 4 h and 24 h in the hippocampus, respectively. Such discrepancy in terms of BDNF response between brain cognitive structures was already reported following mild-intensity EX in rats [47] but could also be explained by the isoflurane anesthesia protocol that was shown to inhibit cutaneous afferents and nociceptive signals [48,49] responsible for the activation of the prefrontal cortex, as reported in humans during EMS [50]. However, the findings on the hippocampus are consistent with previous studies in rats using functional electrical stimulation (FES) mimicking the walking pattern [51], sciatic nerve stimulation [52], or stimulation applied on the biceps and triceps brachii muscles [53], all of which reported hippocampal BDNF upregulation. In addition, our results are in line with data obtained from acute classical EX studies showing an increase in *bdnf* mRNA 2 and 6 h after a single treadmill session [54]. Moreover, the hippocampal BDNF increase was associated with synaptic protein upregulation, since both PSD-95 and GAP-43 were upregulated, without any change concerning SYP. Taken together, our data and the results obtained by others show that an acute EMS protocol can reproduce the results obtained after acute EX in terms of hippocampal BDNF synthesis.

To elucidate the mechanisms underlying the hippocampal BDNF increase after EMS, we focused on the three main pathways involved in this brain neurotrophin overexpression after EX. We first investigated neuronal activity and cerebral blood flow increases, since the activation of these two pathways has been extensively demonstrated after classical EX [5,47,55]. Using c-fos as an indicator of neuronal response [56] and the phosphorylated form of eNOS at serin 1177 as a marker of increased cerebral blood flow [57], we were unable to report an effect of EMS on these two pathways, since no variation in the expression of these markers could be found either 4 or 24 h after EMS. Furthermore, although we did not assess this aspect in our human study, neuroimaging approaches reported an activation of the sensorimotor cortex in response to EMS [58,59], without referencing any activation in regions associated with cognition such as the hippocampus, while in contrast, EX was shown to lead to the activation of the hippocampal region up to 20 min after its completion in humans [60].

We then explored the hypothesis that locally evoked muscle contraction may be responsible for the hippocampal BDNF increase by promoting the synthesis and release of myokines into the bloodstream. We first assessed whether the quadriceps responded to EMS through an increase in BDNF and FNDC5/irisin expression. Indeed, BDNF is produced in skeletal muscle by various cell types [32,61] and was reported to play a role in muscle regeneration and metabolism [62,63,64]. Recent evidence also showed that skeletal muscle could secrete BDNF, which acts as an endocrine myokine to regulate glucose homeostasis by influencing the pancreas [31]. In the brain, peripheral delivery of BDNF was reported to induce neurogenesis and increase the BDNF levels in the hippocampus [65]. Moreover, a significant increase in *bdnf* mRNA and/or BDNF protein expression was observed in skeletal muscle in response to EX [66], while recent evidence showed that EMS increased the blood BDNF levels in humans [67,68] and animals [53]. However, in our experimental settings, we did not observe any change in the quadriceps expression or circulating levels of this protein in response to EMS, ruling out the possibility that muscle-derived BDNF could be responsible for the observed hippocampal BDNF increase. We next focused our analysis on FNDC5/irisin, since compelling evidence suggests that the activation of this pathway within skeletal muscles is involved in EX-induced cerebral plasticity [13]. Although our results showed that quadriceps FNDC5/irisin protein expression was significantly increased 24 h after EMS, ELISA and immunoprecipitation measurements at 0 h, 4 h, and 24 h failed to reveal changes in serum irisin levels. This result points out the fact that the synthesis of FNDC5 must be distinguished from its cleavage, which depends on the activation of a still unknown protease. Thus, it is likely that certain types of EX lead to an increase in FNDC5 synthesis without, however, stimulating the secretase required to produce circulating irisin. In addition to its peripheral expression, FNDC5/irisin is also produced in different brain regions; therefore, we investigated whether EMS could impact its expression within the hippocampus. Regarding the central expression, we did not observe any significant variation in FNDC5/irisin levels in the hippocampus. To complete our analysis, and since irisin was shown to signal through its binding to αVβ5 integrin-type receptors and the induction of downstream signaling pathways involving FAK [34], we further analyzed the effect of EMS on FAK activation. No variation in the expression of phosphorylated FAK could be found either 4 h or 24 h after EMS. Overall, these data, together with the lack of association between muscle FNDC5/irisin expression and hippocampal BDNF expression, seem to exclude both central and peripheral irisin function as a mechanism inducing increased hippocampal BDNF expression in our experimental conditions.

Finally, we turned our attention to lactate, a metabolic myokine, since it was shown to be related to brain BDNF synthesis [20]. As reported in humans, our results showed that our EMS protocol induced a massive and similar rise in lactatemia in rats. Interestingly, in an elegant study using a monocarboxylate transporter antagonist (AR-C155858) that prevents lactate entry into the brain, Hayek et al. demonstrated a complete abolition of hippocampal *bdnf* gene expression in response to EX, whereas intraperitoneal administration of lactate in mice resulted in a hippocampal BDNF increase comparable to that observed in trained mice. Mechanistically, lactate production was shown to mediate the effects of EX on learning and memory through the SIRT1-dependent activation of hippocampal BDNF [20]. Consistently, in our study, we were able to report an increase in SIRT1 expression 4 h after EMS. In addition, beyond its role in inducing SIRT1, lactate could facilitate BDNF expression through the facilitation of NMDA receptor activation [69] or through histone lactylation, which was recently shown to impact gene expression [70], although such a mechanism has not been demonstrated at the BDNF promoter level yet. It is important to highlight that, in our study, the effects of EMS on BDNF expression in rats were positively correlated with blood lactate production. Furthermore, we also found a positive trend of association between the congruent and the incongruent Stroop tasks, whereas a significant inverse correlation was found between anxiety and lactatemia in humans. These outcomes align with a prior investigation showing that the positive effects of EX on executive function were linked to the cerebral arteriovenous difference in lactate [21] and identifies lactate as the molecule underlying the cognitive improvements observed in response to EMS. Although association analyses cannot be interpreted as causal evidence, taken together, our data nevertheless argue in favor of blood lactate as a mediating link in the effect of EMS on BDNF-dependent plasticity in rats and in the EMS-associated promotion of cerebral benefits in humans.

## 4. Materials and Methods

### 4.1. Human and Experimental Design

For studies involving human subjects, the investigations were approved by the scientific committee of the Faculty of Sports Sciences of the University of Burgundy in Dijon (CPP EST: approval number A00064-49) and were conducted in accordance with the principles of the Helsinki Declaration. The participants were fully informed about the study details, potential risks, and discomfort associated with the experiments. All participants provided informed consent before participating.

To implement the EMS, the participants were asked to sit comfortably in a chair equipped with ankle fixation devices to engage the quadriceps muscles isometrically. Two square electrodes measuring 5 × 5 cm (Fyzéa, BAS95050, La Roche-sur-Yon, France) were placed on the vastus lateralis and vastus medialis muscles, while a larger electrode measuring 5 × 10 cm was applied to the rectus femoris muscle to induce quadriceps contraction. These electrodes were connected to a stimulator (Cefar Rehab X2, 111126) that allowed configuring the current parameters as follows: frequency of 100 Hz, pulse width of 0.4 ms, and 7 s ON followed by 14 s OFF, for a duration of 30 min. During the session, the stimulation intensity was gradually increased to reach the maximum tolerable level for each participant (starting from 6 mA and ranging from 26 to 99 mA depending on the subject).

The characteristics of the participants are detailed in Table 1. Participants with neurological, psychiatric, cardiovascular, and metabolic diseases, as well as those taking medications that could interfere with our measurements, were excluded. Based on our inclusion criteria, we included individuals with a BMI ranging from 18 to 25. In a first study involving 40 participants (cohort 1), we investigated the acute effects of EMS on cognitive functions. In this investigation, all subjects visited the laboratory twice, with a 7-day interval between their visits. The first visit aimed to familiarize the participants with the experimenter, the EMS protocol, and the cognitive assessments, while the second aimed to assess the impact of EMS on cognitive abilities. Additionally, we also recorded physiological parameters during EMS sessions, as described below. In a second series of experiments conducted with a group of 8 participants (cohort 2), we replicated the same protocols used for the first cohort, adding the measurement of the lactate levels.

### 4.2. Physiological Recording in Humans

During the 30 min EMS session, we recorded the heart rate (HR) and blood pressure (BP) using a blood pressure monitor (BP 3NZ1-3P, Torm Copenhagen, Denmark), oxygen saturation (SpO2) using a pulse oximeter (SaO2, YK-81CEU, Braun, Kronberg im taunus Germany), as well as changes in body temperature using a forehead laser thermometer (T°, FH2—Thomson, Thermo, Issy-les-Moulineaux, France).

### 4.3. Neuropsychological Assessment in Humans

To assess the effects of EMS on cognition, we conducted tests to evaluate executive function (Stroop test), episodic memory (15-word list), and spatial memory (Rey’s figure). Additionally, we also examined the effects of EMS on self-reported mood states using the POMS (Profile of Mood States) questionnaire.

For the Stroop task, we used 3 pages, each containing 100 items arranged in 10 columns of 10 items. On the first page, the words “bleu” (blue), “rouge” (red), “jaune” (yellow), and “vert” (green) were printed in black ink. On the second page, the words were printed as “XXXX” in blue, red, yellow, and green. The last page reported the same color words printed in a mismatched color (e.g., “bleu” written in yellow ink), and the participant had to name the ink color. The number of words read in 45 s and the number of errors made were assessed. During the familiarization session (7 days before the evaluation session), the participants completed 3 blocks of 45 s for each page to avoid a learning effect. For the evaluation session, the participants completed one block for each condition before and after EMS.

The Rey–Osterrieth complex figure test consisted in copying a complex figure before the stimulation and in a delayed recall of it immediately after the stimulation. Scoring was measured in a double-blind fashion on 72 points.

The 15-word list test was conducted immediately after EMS. The participants were exposed to a first series of 15 words (list A) for a period of 3 min and then were asked to recall as many words as possible. Subsequently, the subjects were presented with a second series of 15 words (list B) for the same duration of 3 min. They were then asked to recall the words from list B and then the words from list A, to assess memory in the presence of interference. Finally, 24 h after the session, the participants were contacted by phone for a delayed recall of list A. The word lists A and B were the same for all subjects.

Regarding the POMS questionnaire, the participants were required to indicate their feelings in relation to 35 adjectives over the past few hours, including the present moment, using a five-point scale ranging from 0 (not at all) to 4 (extremely). The questionnaire is divided into 6 mood categories: anxiety, depression, anger, vigor, fatigue, and confusion. The POMS was administered both before and after the EMS session.

### 4.4. Animals and Experimental Design

The procedures involving animals were conducted in accordance with the guidelines of the French Ministry of Agriculture (license 21-CAE-102) and of Research (APAFIS number #333000) and were approved by the local animal ethics committee in Dijon (approval number 105). Our experimental protocols were carefully designed to minimize any potential suffering in the animals used.

All our studies were conducted on adult male Wistar rats, purchased from Janvier Labs (Le Genest-Saint-Isle, France). The animals were housed in groups of five per cage, with a 12 h light/dark cycle and free access to food and water. As for the housing conditions, the cages were maintained in a ventilated cabinet with a controlled ambient temperature between 20 and 24 °C and humidity ranging from 45 to 65%.

For the EMS protocol, the rats were first subjected to gas anesthesia using isoflurane (5% for induction, followed by 2.5% for anesthesia maintenance). Subsequently, their thighs were shaved to place the stimulation electrodes. The electrodes used were the same as those employed in the human studies but were modified to adapt to the rat morphology. One electrode was secured to the upper part of the quadriceps, while another was positioned on the lower part to trigger muscle contraction. The electrical stimulator used was identical to the one used for humans (Cefar Rehab X2, 111126, Lyon, France), and the current parameters were also the same, i.e., frequency of 100 Hz and pulse width of 0.4 ms, with cycles of 7 s of stimulation followed by 14 s of rest, for a total duration of 30 min. The stimulation intensity was gradually increased, ranging from 6 to 20 mA for each rat to maintain a consistent level of contraction throughout the session.

After a 7-day habituation period to the experimenter, all rats underwent a first habituation session to the EMS protocol. Then, the animals were subjected to a second EMS session 7 days later. At the end of this second EMS session, the rats were sacrificed 4 (n = 16) or 24 h (n = 14) after EMS. The SHAM rats underwent the same procedure in parallel without electrical stimulation.

The rats were euthanized under isoflurane inhalation (5%) by intracardiac infusion of saline (0.9% NaCl). The quadriceps muscle, prefrontal cortex, and hippocampus were dissected and harvested. The right quadriceps were placed in paraformaldehyde (4%, PFA) for histological analysis. The left quadriceps, the left prefrontal cortex, and the left hippocampus were immediately frozen in liquid nitrogen and stored at −80 °C for biochemical analysis. Blood was collected intracardially before saline infusion or by a cut in the tail, centrifuged at 2000× *g* during 15 min at 4 °C to collect the serum, and immediately stored at −80 °C for further analysis.

### 4.5. Quantitative Real-Time PCR

Total RNA from the prefrontal cortex and hippocampus was extracted using NucleoSpin^®^ RNA Set for NucleoZol (740406.50, Macherey-Nagel, Hoerdt, France) following the manufacturer’s recommendations. Reverse transcription was performed with an iScript cDNA Synthesis Kit (Bio-Rad, Hercules, CA, USA). PCR was carried out using Powerup SYBR Green master mix (Applied Biosystems, Life Technologies, Waltham, MA, USA) on a StepOnePlus™ Real-Time PCR System (Applied Biosystems). The relative gene expression was determined using ΔΔCt values. The target mRNA levels were normalized to the β-actin mRNA and 18 S mRNA levels. Primers (Table 2) were purchased from Thermo Fischer Scientific (Waltham, MA, USA).

### 4.6. Western Blotting

Protein extraction was performed using the Precellys system at 4 °C (Berlin Technologies) with 7 volumes of lysis buffer (100 mM Tris base, 150 mM NaCl, 1 mM EGTA, 1% triton X100, protease and phosphatase inhibitors, pH 7.4). After ultrasonic exposure during 15 s, the homogenates were centrifuged at 15,000× *g* for 20 min at 4 °C. The protein content in the supernatant was determined using the Lowry method (Lowry Pierce™, Thermo Fischer Scientific). Aliquots of supernatant were mixed with 2× Laemmli sample buffer (Tris 125 mM, SDS 4%, glycerol 20%, bromophenol blue 0.01%) and stored at −80 °C.

The protein samples were separated by SDS-PAGE electrophoresis on stain-free gels (TGX Stain-Free FastCast Acrylamide kit, Biorad, Hercules, CA, USA) and then transferred to a polyvinylidene fluoride (PVDF) membrane (Biorad) or a nitrocellulose membrane (Biorad) using the TransBlot transfer system (Biorad). A stain-free blot image was acquired using the ChemiDoc imaging system (Biorad) for total protein measurement in each sample lane. The membranes were blocked with 5% non-fat milk during 1 h at room temperature (RT) and then probed overnight at 4 °C with the primary antibodies. The complete list of antibodies used for Western blotting is provided in Table 3. Next, the membranes were incubated with HRP-conjugated secondary anti-rabbit (111-035-144, Jackson ImmunoResearch, Cambridgeshire, UK) or anti-mouse (115-035-166, Jackson ImmunoResearch) antibodies for 1 h at RT. The membranes were revealed using an enhanced chemiluminescence substrate (Biorad, Clarity ECL substrate, 170-5060) and scanned using the ChemiDoc imaging system (Biorad). Band intensity was analyzed using ImageLab software (Version 6.0.1, Biorad) using stain-free blot, β-actin, or αSMA as loading control.

### 4.7. ELISA

ELISA kits were used to measure the levels of irisin (Catalog No. EK-067-29, Phoenix Pharmaceuticals, Mannheim, Germany) and BDNF (BEK-2211-2P, Biosensis, Thebarton, South Australia) in the serum of SHAM and EMS rats, following the supplier’s recommendations. The assays were performed in duplicate. The results were considered valid if the coefficients of variation were below 10% and if the positive control result fell within the range indicated by the supplier.

### 4.8. Immunoprecipitation

We used immunoprecipitation to purify and amplify the signal of irisin in our Western blotting experiments conducted on rat serum. To do this, we utilized protein A-conjugated magnetic beads (Dynabeads, 10002D, Thermo Fisher Scientific, Waltham, MA, USA). These beads were incubated with the anti-FNDC5 antibody (Table 3) for 10 min at RT using a sample mixer (HulaMixer™, 15920D, Thermo Fisher Scientific). Subsequently, the bead–antibody complex was isolated using an appropriate magnet (DynaMag™-2, 12321D, Thermo Fisher Scientific). The rat sera were pooled according to their respective groups and then incubated with the bead–antibody complex for 20 min at RT using a sample mixer. After three washes with phosphate-buffered saline (PBS) solution, the bead–antibody–antigen complex was resuspended in 20 µL of Laemmli 2X buffer and eluted by heating at 95 °C for 10 min. This step released the antibody–antigen complex from the magnetic beads. Finally, the sample was brought into contact with the magnet, and the resulting supernatant was loaded onto the Western blotting gel.

### 4.9. Muscle Histology

The muscles, previously fixed in a 4% PFA solution, were embedded in paraffin blocks and cut into 5 µm sections. The sections were deparaffinized and rehydrated through successive baths of xylene, ethanol, and running water. Subsequently, the sections were stained using baths of hematoxylin (Harris Hematoxylin, Leica 3801562E, Wetzlar, Germany) and eosin (Eosin Y, Leica 3801601E).

### 4.10. Blood Lactate Measurement in Animals and Humans

The concentration of lactate in total blood was measured immediately after the completion of the EMS protocol (0 h) using the Lactate Pro II measuring equipment. Blood samples were taken from the fingertip in humans using a lancet or from the tail in rats after making a slight incision.

### 4.11. Data Analysis and Statistics

All statistical analyses were conducted using GraphPad Prism software (Version 8.0.1). The data are presented as means ± standard deviation (SD). Normality was assessed using the Shapiro–Wilk test. When comparing a single variable between SHAM and EMS rats or CTRL and EMS participants, statistical differences were assessed using the Student’s *t*-test or the Mann–Whitney nonparametric test, depending on the normality of the data. For comparisons involving multiple time points between SHAM and EMS rats or CTRL and EMS participants, one-way analysis of variance (ANOVA) was employed. *p*-values less than 0.05 were considered statistically significant.

## 5. Conclusions

Our data show that EMS could be used as an effective alternative to conventional EX for enhancing cognitive function. These new findings are particularly important for patients who are unable to participate in this type of EX, especially those who are bedridden or suffer from pathologies such as obesity, chronic heart failure, chronic obstructive pulmonary disease, and stroke-associated locomotor disorders. In terms of prospects, our data pave the way for further studies to refine the EMS parameters with the aim of applying this protocol chronically to patients whose cognition is affected by the above-mentioned pathological situations. In addition, although future proof-of-concept studies will be necessary, such as blocking the lactate cerebral transporter and/or increasing its blood concentration under normal and EMS conditions, our study points to lactate as a bridging signaling molecule of the muscle–brain dialogue following EMS. Interestingly, the lactate content has the advantage of being easily manipulated compared to that of the other myokines currently identified.

## Figures and Tables

**Figure 1 ijms-25-01883-f001:**
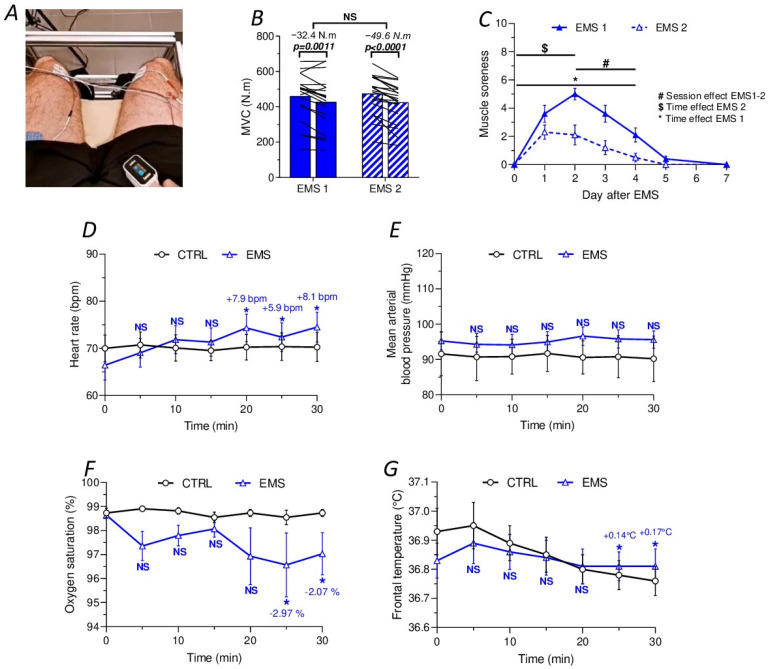
Physiological responses to EMS in humans. (**A**) Representative pictures of electrode placement for EMS. (**B**) Comparison of maximal voluntary contraction (MVC) levels before (left column) and after (right column) the EMS sessions. The left column represents the baseline MVC, while the right column illustrates MVC outcomes following two distinct sessions: a familiarization session (EMS 1, full bar) and a subsequent session (EMS 2, hatched bar). (**C**) Muscle soreness evaluated using a pain scale after the familiarization session (EMS 1, solid line) and the second session (EMS 2, dashed line). (**D**) Heart rate recordings for the control (CTRL) and EMS groups. The differences indicated are in comparison to the baseline values of the EMS group. (**E**) Mean arterial blood pressure derived from systolic blood pressure (SBP) and diastolic blood pressure (DPB) recordings for the CTRL and EMS groups. (**F**) Oxygen saturation recordings obtained from pulse oximeter readings for the CTRL and EMS groups. (**G**) Frontal temperature measured using a laser thermometer in the CTRL and EMS groups. * Different from basal values (*p* < 0.05). NS means no significance.

**Figure 2 ijms-25-01883-f002:**
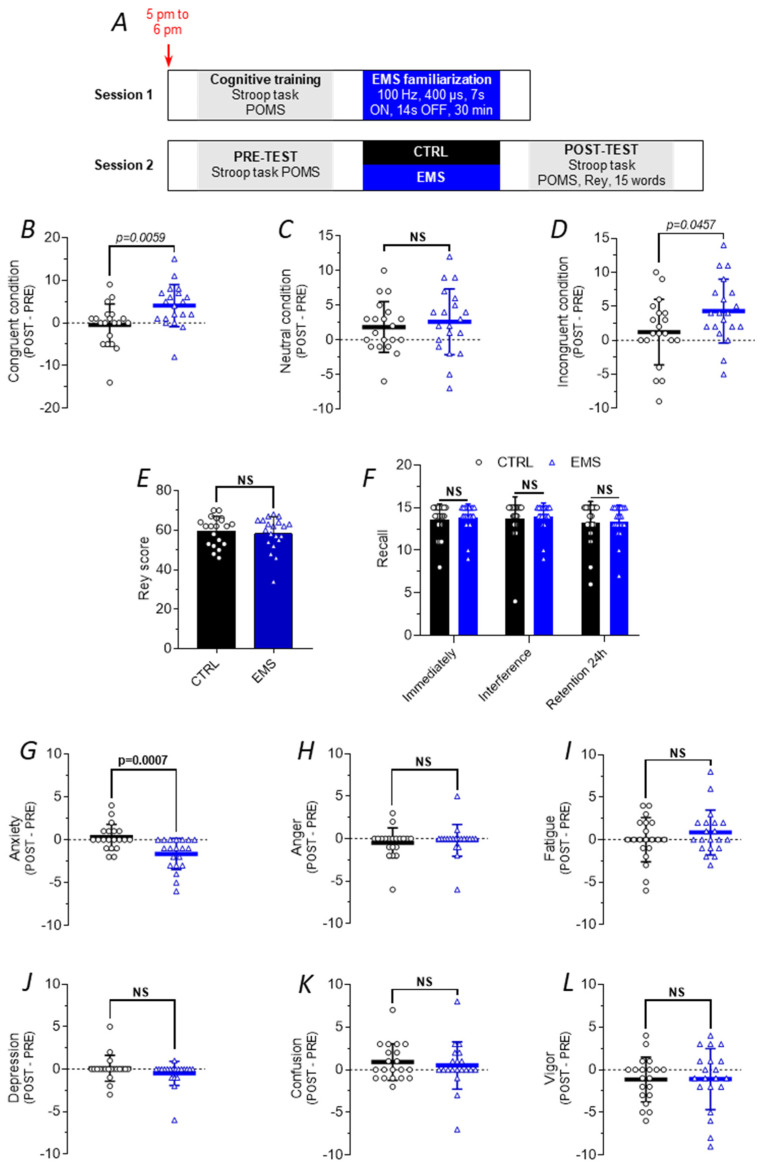
Cognitive and mood responses to EMS in humans. (**A**) Experimental design employed for the human experiment. (**B**–**D**) Changes in the number of words from POST- to PRE-test in the congruent condition of the Stroop task (**B**), the neutral condition (**C**), and the incongruent condition (**D**) following CTRL task completion (black) and EMS session (blue). (**E**) Rey score calculated in a double-blind fashion for both CTRL and EMS groups. (**F**) Number of recalls for the 15-word list immediately following CTRL task completion and EMS, after the use of an interference 15-word list, and 24 h after the experiment. (**G**–**L**) Changes in the profile of mood states, i.e., anxiety (**G**), anger (**H**), fatigue (**I**), depression (**J**), confusion (**K**), and vigor (**L**) following both CTRL task completion (black) and EMS (blue). NS means no significance.

**Figure 3 ijms-25-01883-f003:**
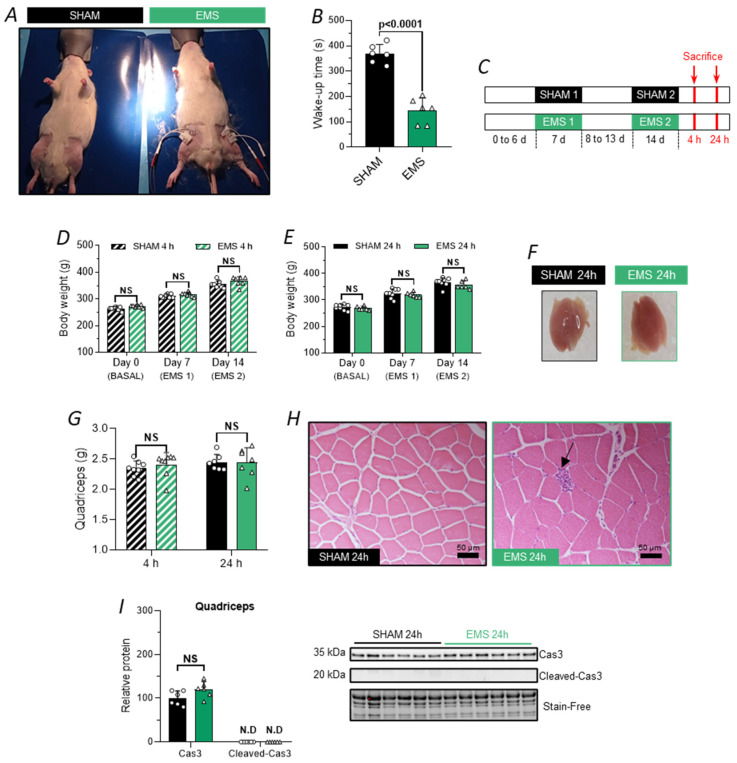
Characterization of the EMS protocol in rats. (**A**) Representative placement of electrodes for the EMS protocol in rats. (**B**) Duration of wake-up time for the SHAM (black bar) and EMS (green bar) groups. (**C**) Experimental design employed in rats, with “d” indicating day, and “h” denoting hour. (**D**,**E**) Changes in body weight of SHAM rats (black bar) and EMS rats (green bar) across the experimental design, 4 h ((**E**), hatched bar) or 24 h ((**F**), full bar) after the application of EMS or control conditions. (**F**). Representative pictures of the quadriceps muscle 24 h after the SHAM or the EMS session. (**G**) Comparison of the quadriceps muscle weight between SHAM (black) and EMS (green) groups. (**H**) Representative photomicrograph of H&E staining of the quadriceps muscle from SHAM and EMS rats 24 h post-EMS. (**I**) Effect of EMS on the relative protein levels of caspase 3 and cleaved form of caspase 3 in the quadriceps muscle 24 h post-EMS. NS means no significance, N.D. means no detected.

**Figure 4 ijms-25-01883-f004:**
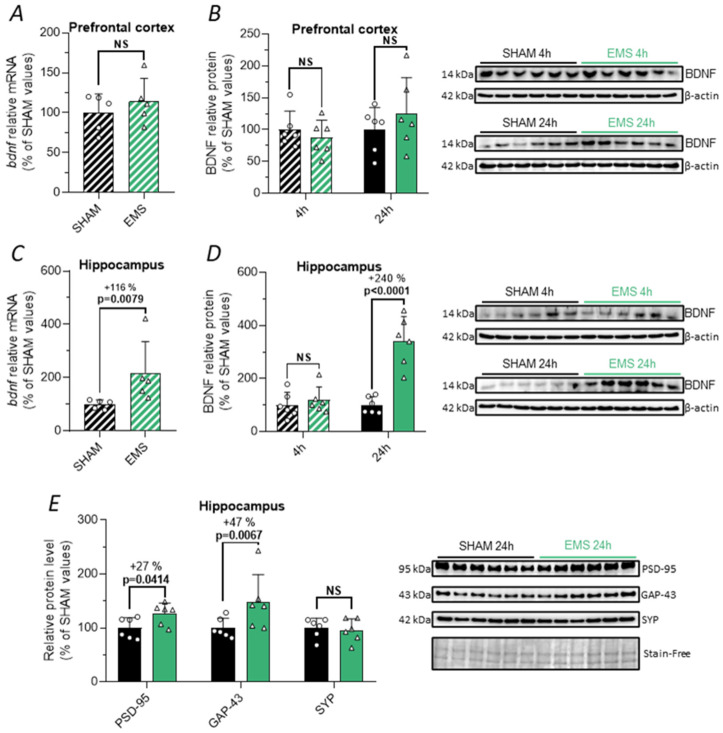
Effect of EMS on BDNF and synaptic protein expression. (**A**–**D**) Relative levels of bdnf mRNA assessed by RT-qPCR in the prefrontal cortex (**A**) and hippocampus (**C**) 4 h after the implementation of the protocols (hatched bar), as well as BDNF relative protein levels in the prefrontal cortex (**B**) and hippocampus (**D**) 4 h (hatched bar) or 24 h (full bar) in the SHAM (black) and EMS (green) groups. (**E**) Relative protein expression of PSD-95, GAP-43, and SYP at the 24 h time point in the SHAM and EMS groups. Corresponding immunoblots are shown on the side of the graphs. NS means no significance.

**Figure 5 ijms-25-01883-f005:**
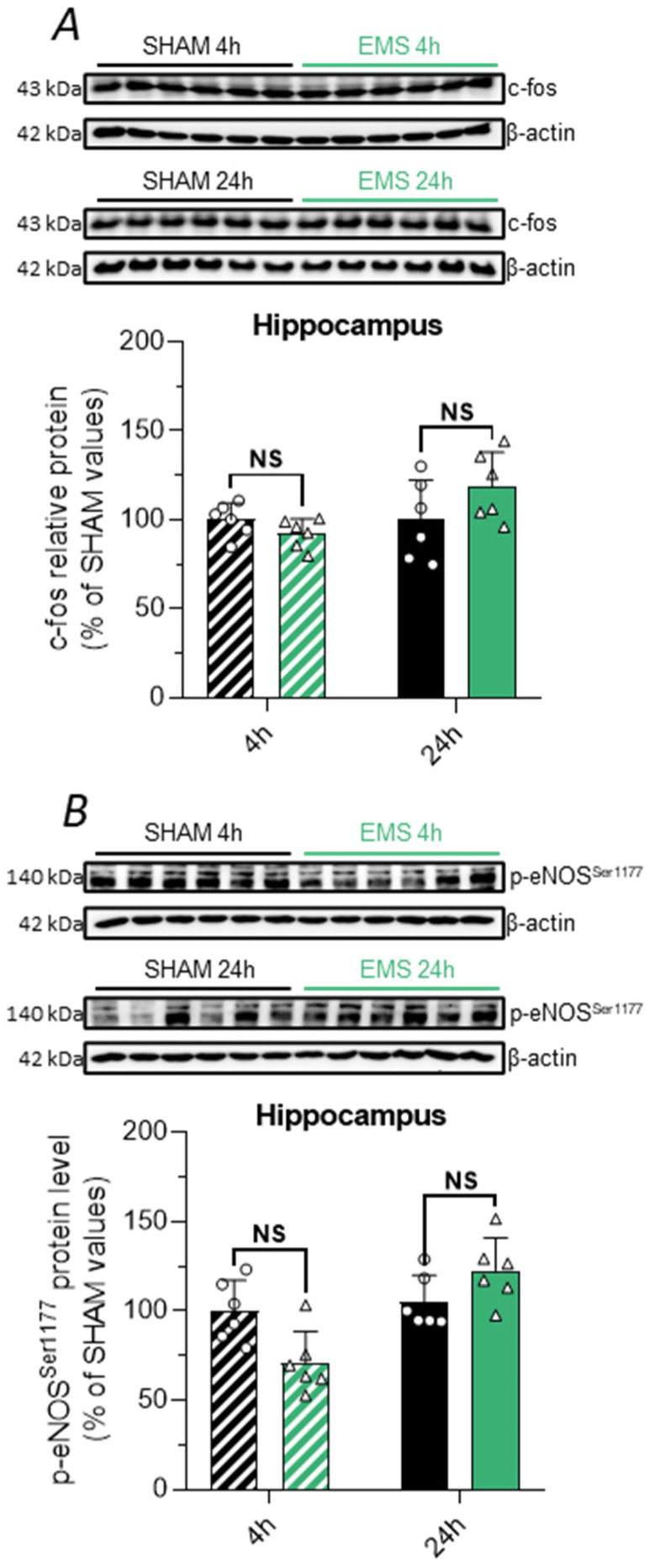
Effect of EMS on neuronal activity and shear stress in rats. (**A**,**B**). Relative protein expression of c-fos (**A**) and p-eNOS^Ser1177^ (**B**) in the rat hippocampus 4 h (hatched bar) or 24 h (full bar) after protocol application in the SHAM (black) and EMS (green) groups. Corresponding immunoblots are shown on the top of the graphs. NS means no significance.

**Figure 6 ijms-25-01883-f006:**
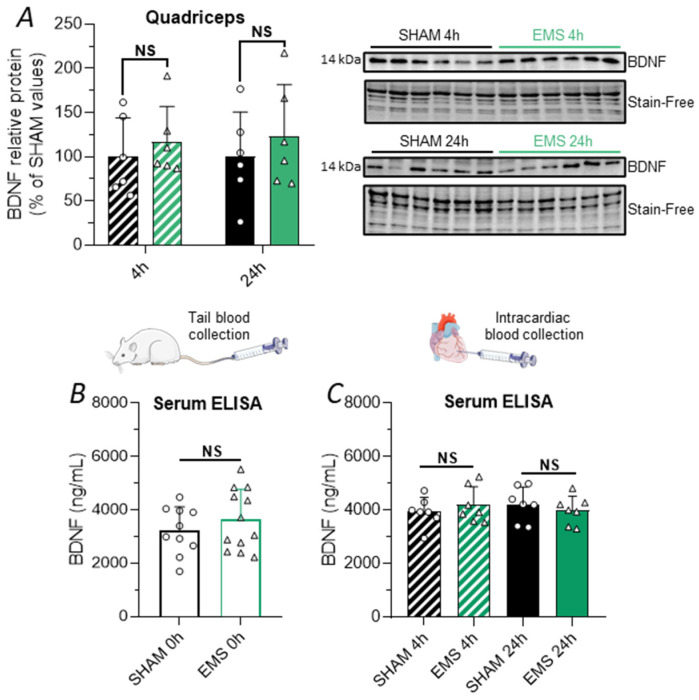
Effect of EMS on muscle-derived BDNF. (**A**) Relative protein expression of BDNF in quadriceps muscle 4 h (hatched bar) or 24 h (full bar) after treatment in the SHAM (black) and EMS (green) groups. Corresponding immunoblots are shown on the side of the graphs. (**B**,**C**) Circulating levels of BDNF measured by ELISA immediately after protocol application and tail blood collection (**B**) and subsequent measurements 4 and 24 h post-EMS, after intracardiac blood collection (**C**). NS means no significance.

**Figure 7 ijms-25-01883-f007:**
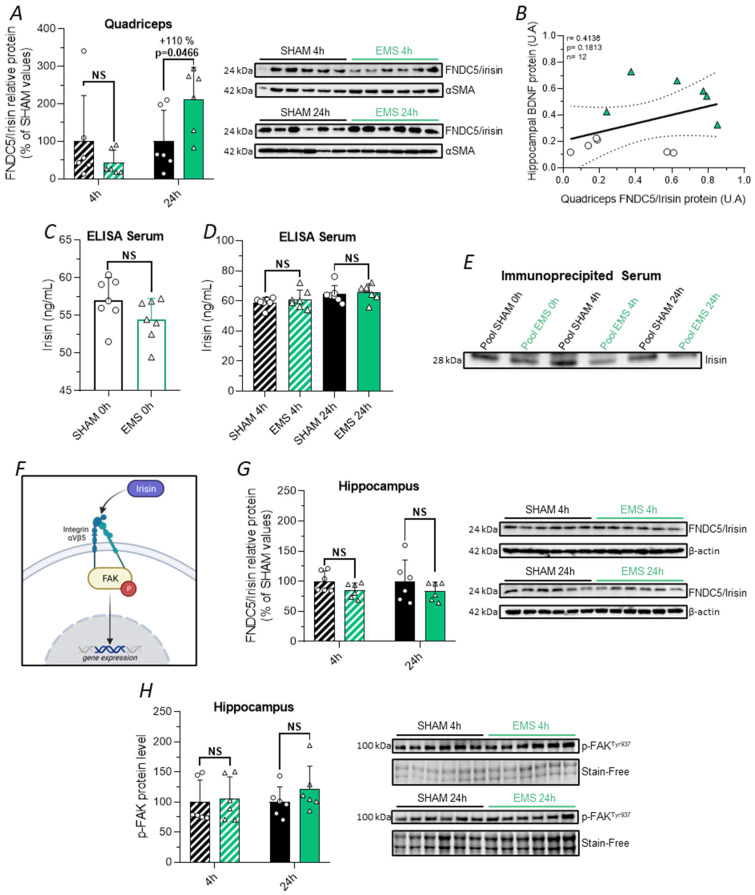
Effect of EMS on the FNDC5/irisin pathway. (**A**) Relative protein levels of FNDC5/irisin in quadriceps muscle at 4 h (hatched bar) and 24 h (full bar) in the SHAM (black) and EMS (green) groups. (**B**) Spearman correlation between hippocampal BDNF and quadriceps muscle FNDC5/irisin relative protein expression levels at the 24 h time point for SHAM rats (black circle) and EMS rats (green triangle). (**C**,**D**) Circulating irisin levels measured by ELISA immediately (tail blood collection, (**C**)) and 4 or 24 h (intracardiac blood collection, (**D**)) after protocol application. (**E**) Representative immunoblots of pooled immunoprecipitated serum immediately, 4 h, and 24 h following the procedures. (**F**) Schematic signaling pathway activated by irisin upon binding to the integrin receptor. (**G**) Relative protein levels of FNDC5/irisin in the hippocampus 4 and 24 h after the procedures. (**H**) Relative protein levels of p-FAK^Tyr397^ in the hippocampus at the 4 h and 24 h time points. Corresponding immunoblots are shown on the side of the graphs. NS means no significance.

**Figure 8 ijms-25-01883-f008:**
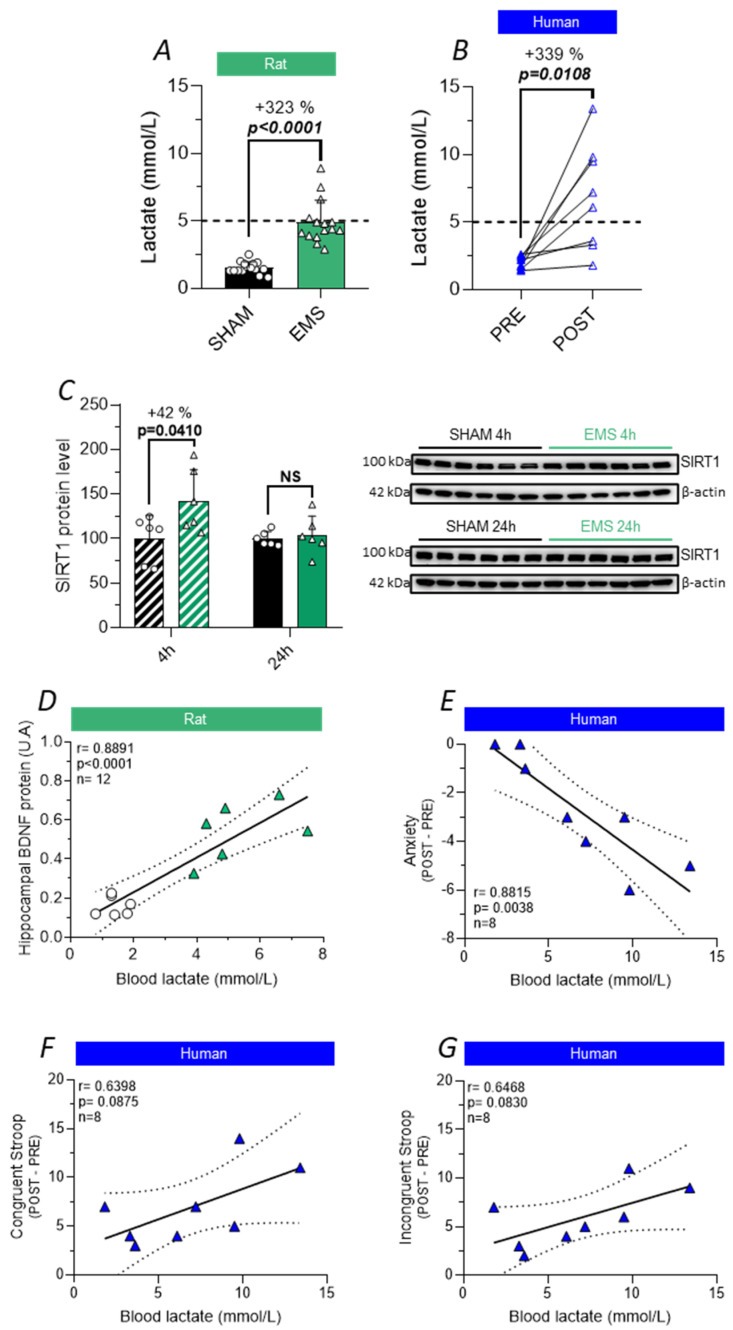
Link between EMS-induced lactate release and hippocampal BDNF expression in rats and cerebral benefits in humans. (**A**,**B**) Changes in lactatemia measured immediately after the procedures in SHAM (black) and EMS (green) rats (**A**) as well as before (full blue triangles) and after (empty blue triangles) the completion of EMS in humans (**B**). (**C**) Relative protein levels of SIRT1 in the hippocampus 4 h (hatched bars) and 24 h (full bars) after the procedures in the SHAM (black) and EMS (green) groups. (**D**) Spearman correlation between hippocampal BDNF relative protein expression at the 24 h time point and lactatemia assessed immediately after the procedures in rats. (**E**–**G**) Spearman correlation between anxiety (**E**), congruent Stroop trial (**F**), incongruent Stroop trial (**G**), and lactatemia in humans. NS means no significance.

**Table 1 ijms-25-01883-t001:** Subjects’ characteristics.

Cohort	Cohort 1		Cohort 2
Group	CTRL	EMS	EMS
Total subjects	20	20	8
Men	14	14	6
Women	6	6	2
Age	22.2 ± 3.4	24.8 ± 2.5	24.2 ± 1.7
BMI	21.7 ± 2.2	21.6 ± 1.9	22.5 ± 3.0
Caffein	8	8	3
Smoke	2	2	1

BMI: Body mass index. Caffeine: number of individuals who consumed caffeine on the day of the experiment. Smoke: number of individuals with regular tobacco consumption.

**Table 2 ijms-25-01883-t002:** Primers used.

Gene Name	Forward	Reverse
*Bdnf* total	*TACCTGGATGCCGCAAACAT*	*TGGCCTTTTGATACCGGGAC*
β-actin	*ATGGAGGGGAATACAGCCC*	*TTCTTTGCAGCTCCTTCGTT*
18S	*GTAACCCGTTGAACCCCATT*	*CCATCCAATCGGTAGTAGCG*

**Table 3 ijms-25-01883-t003:** Primary antibodies.

Protein Name	Reference	Dilution
BDNF	Abcam recombinant anti-BDNF antibody rabbit monoclonal [EPR1292] (ab108319)	1/3000 TBST-Milk (5%)
Cas3	Cell Signaling, D3R6Y, #14220, rabbit mAb	1/3000 TBST-Milk (5%)
c-fos	Genetex rabbit c-fos antibody GTX129846	1/3000 TBST-Milk (5%)
FNDC5/irisin	Abcam recombinant anti-FNDC5 antibody rabbit monocolonal [EPR12209] (ab174833)	1/3000 TBST-Milk (5%)
GAP-43	Cell Signaling GAP43 (D9C8) rabbit mAb (8945)	1/3000 TBST-Milk (5%)
p-eNOS^Ser1177^	BD Transduction Laboratories™ purified mouse anti-eNOS (pS1177) (612392)	1/3000 TBST-BSA (7.5%)
p-FAK^Tyr397^	Cell Signaling phospho-FAK (Tyr397) antibody (3283)	1/3000 TBST-BSA (7.5%)
PSD-95	Cell Signaling PSD95 (D27E11) XP^®^ rabbit mAb (3450)	1/3000 TBST-Milk (5%)
SIRT1	Cell Signaling SirT1 (D1D7) rabbit mAb (9475)	1/3000 TBST-Milk (5%)
SYP	Interchim rabbit polyclonal RB-1461-P1	1/3000 TBST-Milk (5%)
αSMA	Abcam anti-alpha skeletal muscle actin antibody [Alpha Sr-1] (ab28052)	1/3000 TBST-Milk (5%)
β-actin	Sigma-Aldrich monoclonal anti-β-actin (A5441)	1/3000 TBST-Milk (5%)

## Data Availability

The datasets used and/or analyzed during the current study are available from the corresponding author on reasonable request.

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
