# Peer review of "Cerebral Benefits Induced by Electrical Muscle Stimulation: Evidence from a Human and Rat Study"

_ijms, 2024, doi:10.3390/ijms25031883_

Round 1
Reviewer 1 Report
Comments and Suggestions for Authors
The paper explores the muscle electromyostimulation (EMS) as a potential alternative tool for enhancing cognitive function. The topic is of great interest, however, in my opinion the Authors should explain how EMS is capable of such great effects on the cognitive functions after only one session whereas acute physical exercise has only small effects. These results, if confirmed, could suggest that EMS could be more effective than physical exercise in improving cognitive function.
Author Response
Although we're pleased to note that reviewer 1 considers the effects of our EMS protocol to be “great”, these effects are significant but modest compared with what a conventional physical exercise protocol can produce. Indeed, on the Stroop task, as compared to conventional exercise, we did not have an impact on the neutral condition of the Stroop task while a very slight significant effect was obtained in the incongruent condition (+3.1 words, p=0.0457). Regarding the POMS, only anxiety was decreased while following classical EX, several studies have reported that depression, confusion, and anger are reduced after acute EX (examined in Basso et al., 2017, ref 2 of the manuscript). However, we do agree with the reviewer concerning the fact that EMS could be an alternative to classical EX but to confirm this hypothesis, we will compare in the future to what extent the cognitive effects of EMS can be paralleled to those obtained following traditional exercise by including within a same study both an EMS group and an exercise group.
To be consistent with the message we want to get across and to address reviewer 1 remark, although we already stated in the discussion section that EMS only captures a fraction of the benefits that exercise imparts on cognition and mood (between lines 381 and 387), in the salient observations paragraph of the discussion (lines 355 and 356), the terms “although significant but modest compared to the effect of classical exercise” was added.
Reviewer 2 Report
Comments and Suggestions for Authors
In this work, the authors assessed the effect of electromyostimulation on cognition in humans and the expression of BDNF in regions related to cognition in rats (prefrontal cortex, hippocampus). Then they investigated the underlying mechanisms behind the observations. Three mechanisms were proposed including neuronal activity, cerebral shear stress, and the release of BDNF, irisin and lactate by skeletal muscle. Results indicated that lactate could represent an appealing candidate to explain changes in hippocampal BDNF expression in response to the EMS protocol. Conclusions from the study suggest EMS could be used as an effective alternative to conventional EX for enhancing cognitive function especially for patients with pathological situations. Overall the authors conducted a comprehensive analysis on the positive effect of EMS, and the manuscript could be potential interest of IJMS readers.
1. In human studies, physiological conditions can significantly affect the evaluation on cognitive function and mood. Did the authors assess the health/mental conditions of all the participants before the study?
2. For the rat study in figure 3H, what is the potential explanation of the observation of infiltration of inflammatory cells in the EMS group?
Author Response
- In human studies, physiological conditions can significantly affect the evaluation on cognitive function and mood. Did the authors assess the health/mental conditions of all the participants before the study?
We would like to thank reviewer 2 for his pertinent comment. Indeed, health/mental conditions were assessed before the study. Thus, we excluded participants with neurological, psychiatric, cardiovascular, and metabolic diseases. Additionally, we gathered information on participants' medication use and excluded those taking medications that could interfere with our measurements (e.g., anxiolytic medications). Finally, we included only individuals with a BMI ranging from 18 to 25.
This important information is now included in the manuscript's Section 4.1 Human and Experimental Design of the revised version of our manuscript (lines 509 to 512). “Participants with neurological, psychiatric, cardiovascular, and metabolic diseases, as well as those taking medications that could interfere with our measurements, were excluded. Our inclusion criteria were restricted to individuals with a BMI ranging from 18 to 25.”.
- For the rat study in figure 3H, what is the potential explanation of the observation of infiltration of inflammatory cells in the EMS group?
It is documented in the literature that unusual exercises, eccentric exercises, and muscle electrostimulation (EMS) can induce alterations in muscle tissue, manifested by sarcomere tearing, sarcolemma permeabilization, and sometimes muscle fiber death. These alterations are caused by the mechanical tension exerted on the muscle during contraction, disruption of calcium homeostasis activating calpain and phospholipases, and ischemia in muscle cells due to the compression of blood vessels during tetanic contraction (Proske & Morgan, 2001; Gissel, 2005; Stozer et al, 2020). Following these alterations, various types of inflammatory cells invade the skeletal muscle with the purpose of i) phagocytosing and eliminating damaged cells and debris, and ii) secreting mediating molecules capable of inducing the proliferation and differentiation of satellite cells for the repair of muscle tissue. A detailed description of the inflammatory response following exercise is outlined in the review by Peake et al. (2017) although it goes beyond the scope of our article.
In order to address the reviewer 2’s comment, in the results section of the manuscript, (2.3 Characterization of EMS protocol in rat (lines 192 to 203), we added the following sentence: “Since it is documented in the literature that EMS can induce alterations in muscle tissue [29], manifested by sarcomere tearing, sarcolemma permeabilization, and sometimes muscle fiber death, we conducted H&E staining of the quadriceps muscles to assess potential infiltration of inflammatory cells. As previously described in humans, and with the purpose of phagocytosing damaged cells and debris, and secreting mediating molecules capable of inducing the proliferation and differentiation of satellite cells for the repair of muscle tissue, we observed a rare infiltration of inflammatory cells (Figure 3H). As apoptosis has been detected in muscular diseases and is involved in myofiber cell death, we also assessed caspase 3 and cleaved caspase-3 expression [30]. The modest alteration of muscle tissue was confirmed by the absence of an increase of caspase-3 expression and the undetectable levels of its cleaved form (activated) in quadriceps muscle tissue at 24 hours’ time-point (Figure 3I).” Accordingly, this complementary result is now included as figure 3I in the revised version of the manuscript.
Proske U & Morgan DL (2001) Muscle damage from eccentric exercise: mechanism, mechanical signs, adaptation and clinical applications. J Physiol 537: 333–345.
Gissel H (2005) The role of Ca2+ in muscle cell damage. Ann N Y Acad Sci 1066: 166–180.
STOŽER A, VODOPIVC P & KRIŽANČIĆ BOMBEK L (2020) Pathophysiology of Exercise-Induced Muscle Damage and Its Structural, Functional, Metabolic, and Clinical Consequences. Physiol Res 69: 565–598.
Peake JM, Neubauer O, Della Gatta PA, Nosaka K. 2017. Muscle damage and inflammation during recovery from exercise. Journal of Applied Physiology 122:559–570. doi:10.1152/japplphysiol.00971.2016